# Diterpenoids with Potent Anti-Psoriasis Activity from *Euphorbia helioscopia* L.

**DOI:** 10.3390/molecules29174104

**Published:** 2024-08-29

**Authors:** Zhen-Zhu Zhao, Xu-Bo Liang, Hong-Juan He, Gui-Min Xue, Yan-Jun Sun, Hui Chen, Yin-Sheng Zhao, Li-Na Bian, Wei-Sheng Feng, Xiao-Ke Zheng

**Affiliations:** 1School of Pharmacy, Henan University of Chinese Medicine, Zhengzhou 450046, China; zzhenzhu0921@163.com (Z.-Z.Z.); 15838258630@163.com (X.-B.L.); xueguimin123@126.com (G.-M.X.); sunyanjun2011@hactcm.edu.cn (Y.-J.S.); chenhuiyxy@hactcm.edu.cn (H.C.); zhaoyinsheng101410@163.com (Y.-S.Z.); 1828179389@qq.com (L.-N.B.); 2Academy of Chinese Medical Sciences, Henan University of Chinese Medicine, Zhengzhou 450046, China; 543894206@qq.com

**Keywords:** *Euphorbia helioscopia* L., Euphorbiaceae, chemical investigation, diterpenoids, structure elucidation, anti-proliferative activities

## Abstract

Psoriasis, an immune-mediated inflammatory skin disorder, seriously affects the quality of life of nearly four percent of the world population. *Euphorbia helioscopia* L. is the monarch constituent of Chinese ZeQi powder preparation for psoriasis, so it is necessary to illustrate its active ingredients. Thus, twenty-three diterpenoids, including seven new ones, were isolated from the whole herb of *E. helioscopia* L. Compounds **1** and **2**, each featuring a 2,3-dicarboxylic functionality, are the first examples in the *ent*-2,3-*sceo*-atisane or the *ent*-2,3-*sceo*-abietane family. Extensive spectroscopic analysis (1D, 2D NMR, and HRMS data) and computational methods were used to confirm their structures and absolute configurations. According to the previous study and NMR data from the jatropha diterpenes obtained in this study, some efficient ^1^H NMR spectroscopic rules for assigning the relative configurations of 3*α*-benzyloxy-jatroph-11*E*-ene and 7,8-*seco*-3*α*-benzyloxy-jatropha-11*E*-ene were summarized. Moreover, the hyperproliferation of T cells and keratinocytes is considered a key pathophysiology of psoriasis. Anti-proliferative activities against induced T/B lymphocytes and HaCaT cells were tested, and IC_50_ values of some compounds ranged from 6.7 to 31.5 μM. Compounds **7** and **11** reduced the secretions of IFN-*γ* and IL-2 significantly. Further immunofluorescence experiments and a docking study with NF-*κ*B P65 showed that compound **13** interfered with the proliferation of HaCaT cells by inhibiting the NF-*κ*B P65 phosphorylation at the protein level.

## 1. Introduction

Psoriasis is a common autoimmune skin disease characterized by T cell-mediated hyperproliferation of keratinocytes. This disease has certain distinct but overlapping clinical phenotypes, including chronic plaque lesions (psoriasis vulgaris), acute and usually self-limiting guttate-type eruptions, seborrhoeic psoriasis, pustular lesions, and at least 10% of these patients develop arthritis [1,2]. The global prevalence of the disease is approximately 2%, with regional variations [2,3]. Biological agents, immunosuppressants derived from natural products, and traditional Chinese medicine formulas are the familiar drugs for treating psoriasis. However, the emergence of drug resistance necessitates the discovery of more potent agents for psoriasis treatment.

*Euphorbia*, the largest genus of Euphorbiaceae, contains over 2000 species. Its members inhabit worldwide, some of which are used as folk medicines in China to treat skin diseases (such as psoriasis), edema, tuberculosis, and constipation. The chemical constituents of *Euphorbia* are terpenes, flavonoids, tannins, and phenolic compounds. The characteristic components are diterpenoids, more than 150 diterpenoids with various skeletons and significant bioactivities, such as antitumor, anti-inflammatory, and anti-HIV activities, having been isolated from this genus [4,5]. Diterpenoids from *Euphorbia peplus* (pepluacetal and pepluanols A-B), *E. helioscopia* (helioscopids A and O, euphohelioscoids A), and *E. fischeriana* inhibit Kv1.3 ion channel, a validated target for the treatment of autoimmune diseases, such as multiple sclerosis, type-1 diabetes, asthma, and psoriasis [6,7,8,9]. Additionally, it was reported that the methanol extract of *E. kansui* radix alleviated the symptoms of psoriasis through the inhibition of Th17 differentiation and activation of dendritic cells. These effects are expected to be beneficial in treating and preventing psoriasis [10].

*Euphorbia helioscopia* L., commonly referred as Zeqi in traditional Chinese medicine, is a representative herb of the *Euphorbia* genus. In recent years, the chemical analysis of *E. helioscopia* L. has yielded novel diterpenoids with diverse biological activities, such as secoheliosphane B with activity against HSV-1, heliojatrone C with anti-inflammatory activity, euphohelioscopoids A–C against paclitaxel-resistant A549 human lung cancer cell line, and helioscopids A/O/euphohelioscoids A with Kv1.3 inhibitory activity [7,8,11,12,13], which, in turn, sustains extensive attention from phytochemists. Additionally, multiple clinical studies have found that Chinese ZeQi powder preparation can treat psoriasis without significant adverse reactions [5,14], and *E. helioscopia* L. is the monarch constituent. To find more diterpenoids with promising activities for psoriasis, *E. helioscopia* L. was selected to be investigated in this study, resulting in the isolation of 23 diterpenoids. Among them, compounds **1** and **2** each feature an unusual 2,3-*seco ent*-2,3-dicarboxyl-atisan-2,3-dioic acid or 2,3-*seco ent*-abietan-2,3-dioic acid and compounds **3**–**7** are new jatropha diterpenes. In establishing the structures of new and known jatropha diterpenes, we summarized and examined the correlations between ^1^H NMR signals (chemical shifts and coupling constants) and the relative configurations of C-2, C-13, and C-14 in the (7,8-*seco*)/3*α*-benzyloxy-jatropha-11*E*-ene systems. Here, the isolation and structural elucidation of compounds were reported. Moreover, the anti-psoriatic potential of the isolated compounds was evaluated by assessing their immunosuppressive activities and anti-proliferation effects on keratinocytes. The possible mechanism of active compounds was explored through immunofluorescence experiments and docking studies.

## 2. Results

### 2.1. Structure Elucidation

Compound **1**, isolated as white amorphous powder, has a molecular formula C_20_H_28_O_5_ deduced by HRESIMS ion peak at *m*/*z* 371.1808 [M+Na]^+^ (calcd for C_20_H_28_O_5_Na, 371.1829), appropriate for seven degrees of unsaturation. The absorptions at 1701 and 1721 cm^−1^ in the IR spectrum implied the presence of carboxyls. ^1^H NMR spectrum showed typical signals for a terminal olefine [*δ*_H_ 4.67 (s), 4.88 (s) H_2_-17), and three methyl singlets (*δ*_H_ 0.78, H_3_-20; 1.12, H_3_-19; 1.21, H_3_-18). ^13^C and DEPT NMR data displayed twenty carbons corresponding to three methyls (*δ*_C_ 17.4, C-20; 21.7, C-19; 29.5, C-18), six sp^3^ methenes (*δ*_C_ 19.5 C-6; 28.1, C-11; 31.2, C-7; 39.0, C-1; 42.9, C-15; 44.7, C-13), four sp^3^ methines (*δ*_C_ 38.6, C-12; 45.9, C-9; 48.3, C-5), two sp^3^ quaternary carbons (*δ*_C_ 41.3, C-10; *δ*_C_ 45.5, C-4; 47.7, C-8), a disubstituted terminal double bond (*δ*_C_ 107.3, C-17; 147.0, C-16), two carboxyl groups (*δ*_C_ 178.7, C-2; 187.5, C-3), and one ketone (*δ*_C_ 217.2, C-14) (Table 1). These data from **1** shared a certain similarity to the co-isolate, *ent*-16*β*,17-dihydroxyatisan-3-one (**22**) (Figure 1) [15]. However, as four degrees of the unsaturation accounted for two carboxyls, one double bond, and a ketone, the remaining three degrees of unsaturation required compound **1** to have a tricyclic system, so compound **1** may be a novel *ent*-*seco-*atisane diterpenoid. To verify this deduction and establish the specific structure of **1**, a detailed analysis of ^1^H-^1^H COSY and HMBC correlations was proceeded subsequently (Figure 2). B/C/D rings with a double bond at C16-C17 and a ketone at C-14 were successfully constructed by the ^1^H-^1^H COSY correlations of H-5 (*δ*_H_ 2.54)/H_2_-6a (*δ*_H_ 1.61)/H_2_-7a (*δ*_H_ 2.32), and H-9 (*δ*_H_ 2.69)/H_2_-11 (*δ*_H_ 1.82, 1.56)/H-12 (*δ*_H_ 2.74)/H_2_-13 (*δ*_H_ 2.31), together with the HMBC correlations from H_2_-7 (*δ*_H_ 2.32, 0.98) to C-8 (*δ*_C_ 47.7), H_2_-17 to C-12 (*δ*_C_ 38.6)/C-15 (*δ*_C_ 42.9), H-12 to C-14, and H_3_-20 to C-5 (*δ*_C_ 48.3)/C-9 (*δ*_C_ 45.9)/C-10 (*δ*_C_ 41.3). For ring A, observed HMBC correlations from H_3_-20 to C-1, H_2_-1 to a carboxyl (C-2), and H_3_-18/19 to a carboxyl (C-3), together with unobserved correlations from H_2_-1 to C-3, highlighted an oxidative cleavage between C-2 and C-3 (Figure 2). Therefore, the planar structure of **1** was established, as depicted in Figure 1. Combining consideration of the rotating frame Overhauser effect spectroscopy (ROESY) spectrum and biosynthetic way, as well as comparing the experimental CD curve with the calculated ECD trend (Figure 2 and Figure 3) (Appendix A), the absolute structure of **1** was identified as *ent*-2,3-*seco*-14-oxo-16-atisene-2,3-dioic acid.

The molecular formula of compound **2**, C_20_H_26_O_6_, was determined based on HRESIMS and ^13^C NMR spectra, indicating eight degrees of unsaturation. The max absorption of 275.0 (4.54) in the UV spectrum and absorptions of 1735/1674 cm^−1^ in the IR spectrum indicated a large conjugated system and ketone groups in compound **2**, respectively. 1D NMR data and HSQC spectrum displayed twenty carbon resources attributed to four tertiary methyls (*δ*_C/H_ 8.3/1.83, CH_3_-17; 29.5/1.23, CH_3_-18; 21.3/1.20, CH_3_-19; 21.0/1.01, CH_3_-20), four sp^3^methylenes, three sp^3^ methines (one oxygenated, *δ*_H/C_ 4.86/75.6, CH-12), two double bonds [*δ*_C/H_ 155.6, C-13; 116.9, C-15; 151.1, C-8; 114.5/6.30 (s), CH-14], two carboxyls (*δ*_C_ 178.6, C-2; 186.9, C-3) and a lactone carbonyl (*δ*_C_ 175.1, C-16) (Table 1). Signals of a five-membered carbon ring of *α*,*β-*unsaturated lactone, together with similar 1D NMR spectra data (Table 1) to those of helioscopinolide L, a chemical previously isolated from *E. helioscopia* [16,17], indicated that compound **2** possessed an abietanolide skeleton. However, as three carbonyls and two olefinic bonds accounted for the eight degrees of unsaturation, the remaining three degrees of unsaturation suggested that one ring was opened, so **2** is a tricyclic diterpenoid, too. Detailed analysis of the HMBC correlations from H_2_-1a [*δ*_H_ 2.68 (d, *J* = 20.3 H_Z_)] to a carboxyl (C-2) and H_3_-18/H_3_-19 to another carboxyl (C-3) revealed the break of C2-C3 bond in **2** as compound **1**. Therefore, compound **2** is a novel 2,3-*seco*-abietanolide (Figure 2). So far, the planner structure has been constructed, as shown in Figure 1. Given ROESY correlations (Figure 2) and the similar biosynthetic way as other abietanolides from this plant [5,16], the absolute structure of compound **2** was deduced as *ent*-2,3-*seco*-2,3-dicarboxy-abieta-8(14),13(15)-dien-16,12-olide. This deduction was verified by comparing the experimental CD and calculated ECD spectra (Figure 3) (Appendix A).

Based on the HRESIMS result and ^13^C NMR spectrum, the molecular formula of **3** was determined to be the same as the co-isolate secoheliosphane B (**21**) [11], C_31_H_40_O_8_. These 1D NMR data presented a set of rare but typical signals for a *seco*-jatropha triester: four methyl singlets, two methyl doublets, two acetoxys, two double bonds, one benzyloxy, one ketone, and a ketal (Table 2 and Table 3). Data from **3** were almost identical to those of **21**, implying that the planer structure of compound **3** is the same as that of compound **21**, which was supported by ^1^H-^1^H COSY and HMBC correlations (Appendix A). The geometries of ∆^5^ and ∆^11^ double bonds were easily assigned as *E* for the large coupling constant (^3^*J*_H-11_, _H-12_ = 15.6 Hz) and the ROESY correlations (for ∆^5^: H-4/H_3_-17, H-5/H-7; for ∆^11^ H-11/H-13, H_3_-18/H-12) (Appendix A). For all jatropha diterpenes, the orientations of the substitutions are usually fixed at C-3 (an *α*-oriented benzoate substituent), C-4 (a *β*-oriented hydrogen), and C-15 (an *α*-oriented hydroxy or acetoxy), but orientations of CH_3_-16 at C-2, CH_3_-20 at C-13, and OH(OAc)-14 at C-14 are usually changeable. Carefully analysis of ^1^H and ^13^C NMR data for compounds **3** and **21** both measured in CDCl_3_ revealed the relative configurations at C-2 and C-14 may be opposite for the differences observed in chemical shifts of relative carbons [**3**: *δ*_CH-2_ 39.4/2.15 (m), **21**: *δ*_CH-2_ 38.2/2.46 (m); **3**: *δ*_CH-14_ 74.7/5.82 (d, *J* = 9.5 Hz), **21**: *δ*_CH-14_ 75.3/5.59 (d, *J* = 7.2 Hz); **3**: *δ*_CH3-16_ 14.0/0.96 (d, *J* = 7.0 Hz), **21**: *δ*_CH3-16_ 18.2/1.17 (d, *J* = 7.2 Hz)] (Table 2 and Table 3). The relative configurations of C-2 and C-14 in compound **3** were confirmed by the analysis below. A key ROESY correlation (Appendix A) of H-2 (*δ*_H_ 2.15, m) and H-4 (*δ*_H_ 3.61, dd, *J* = 9.4, 4.0 Hz) indicated that the orientation of CH_3_-16 is *α*, leading to the obvious changes for *J* values of H-3 [**3**: *δ*_H_ 5.65 (t, *J* = 4.0 Hz), **21**: *δ*_H_ 4.69 (t, *J* = 9.0 Hz)], and for chemical shifts of C-1, C-2, C-3, and C-16 (**3**: *δ*_C_ 46.8 C-1, 80.7 C-3; **21**: *δ*_C_ 40.8 C-1, 82.5 C-3). Then, a large coupling constant between H-13 and H-14 (*J* = 9.5 Hz) indicated a *trans*-position for H-13 and H-14. Furthermore, the relative configuration of CH_3_-20 was deduced to be *β*-oriented for chemical shifts of H-11 (*δ*_H_ 5.38) larger than H-12 (*δ*_H_ 5.27) [12,13]. Therefore, the structure of **3** was defined as shown in Figure 1 and named secoheliosphane C.

HRESIMS ion peak at *m*/*z* 537.2459 [M+K]^+^ (calcd for 537.2449) and ^13^C NMR spectrum afforded compound **4** a molecular formula, C_29_H_38_O_7_, missing a mass unit of -CH_2_CO- compared with secoheliosphane B (**21**). A detailed analysis of 1D and 2D NMR data (Table 2 and Table 3) showed that the planner structure of compound **4** is also a *seco*-jatropha ester. The only difference between **4** and **21** is that OAc-15 in **21** is changed into OH-15 in **4**, which was supported by ^1^H-^1^H COSY and HMBC correlations (Appendix A). Subsequently, the relative configuration of **4** was determined by multiple methods, including ROESY (Appendix A), *J*-based configuration analysis (JBCA), and a comparison of chemical shifts. First, the geometries of ∆^5^ and ∆^11^ double bonds were *E* based on the ROESY correlations and *J* values. Subsequently, JBCA of H-3 [*δ*_H_ 5.20 (dd, *J* = 8.5, 5.7 Hz)] together with a chemical shift of CH_3_-16 (*δ*_C_ 18.5), suggested a 16*β-*CH_3_; a small coupling constant between H-13 and H-14 (*J* = 3.1 Hz) indicated that CH_3_-20 and OAc-14 located co-facial. Moreover, compared with secoheliosphane A (*δ*_H-11_ 5.41, *δ*_H-12_ 5.46), secoheliosphane B (*δ*_H-11_ 5.46, *δ*_H-12_ 5.37) and compound **3** (*δ*_H-11_ 5.38, *δ*_H-12_ 5.27), for compound **4**, the chemical shift of H-11 (*δ*_H_ 5.58), larger than that of H-12 (*δ*_H_ 5.64) suggests a 20*α*-CH_3_. Finally, the structure of **4** was defined as shown in Figure 1 and given a trivial name, secoheliosphane D.

Based on the known jatropha-skeleton diterpenoids with acetoxy groups from *E. helioscopia* [5], 1D NMR data of compounds **5***–***7** (Table 2 and Table 3) displayed a group of characteristic signals for jatropha ester framework corresponding to one benzyloxy group, one ketone, two methyl doublets, three methyl singlets, two double bonds, and one to three acetoxy groups. In addition, the remaining two degrees of unsaturation for compounds **5***–***7** revealed that they all are dicyclic jatropha esters.

Compound **5**, C_31_H_40_O_8_, has the same molecular formula and closely similar 1D NMR with co-isolate **15** [13], implying that they share an identical planner structure (Table 2 and Table 3). Given JBCA, and the summarized correlations of the chemical shifts of the relevant carbons and the configurations of C-2, C-13, and C-14 (Figure 4 and Table 2) [12], the orientations of CH_3_-16, CH_3_-20, and OAc-14 were respectively deduced as *α*, *α*, *β* for coupling constants [*δ*_H-3_ 5.41 (t-like, *J* = 3.1 Hz), *δ*_H-14_ 5.90 (d, *J* = 10.3 Hz)] and chemical shifts (*δ*_C-16_ 13.9 < 15.0, *δ*_C-13_ 40.9 > 40.0, *δ*_H-11_ 5.15 < *δ*_H-12_ 5.25). Finally, the structure of **5** was established, as shown in Figure 1, and given a trivial name, heliosco-jatrophane A.

The molecular formula of heliosco-jatrophane B (**6**) is C_33_H_42_O_9_, implying that it may be an acetylated derivative of **5**. Then, HMBC correlation from H-7 [*δ*_H_ 5.03 (dd, *J* = 8.1, 1.7 Hz)] to a carbonyl (*δ*_C_ 170.2) suggested that OH-7 in **5** was acetylated in **6** (Appendix A). The relative configuration of **6** is the same as **5** for similar ROESY correlations (Appendix A), coupling constants, and chemical shifts. Consequently, the structure of **6** was established, as shown in Figure 1.

The molecular formula of heliosco-jatrophane C (**7**), C_29_H_38_O_7_, implied one acetoxyl group in **5** was changed into a hydroxy in **7**. In the ^13^C NMR spectrum (Table 3), an obvious difference was the chemical shift of a quaternary carbon (*δ*_C_ 83.6 in **7**; *δ*_C-15_ 90.2 in **5**), indicating that the hydroxy at C-15 was reserved in **7**. Similarly, the orientations of CH_3_-16, OAc-14, and CH_3_-20 were speculated to be *β, α* and *β* by analysis of ROESY spectrum (Appendix A), JBCA [*δ*_H-3_ 5.17 (dd, *J* = 8.8, 3.6 Hz), *δ*_H-14_ (5.14, d, *J* = 9.0 Hz)] and chemical shifts (*δ*_C-16_ 19.2 > 15; *δ*_H-11_ 5.43 > *δ*_H-12_ 5.15).

The structural elucidation of natural products consistently challenges chemists despite the ongoing development of new methods, including applying derivatization with chiral reagents, using JBCA, examining the intensity of the nuclear Overhauser effect, and relying on quantum-chemical calculations or X-ray crystallographic analysis [12,13,18,19]. Among these, the NMR technique is a convenient and useful procedure for determining the relative configurations of organic molecules. For all jatropha diterpenes, the relative configurations of 3*α*-, 7*α*-, and 15*α*-oxygenated functionalities, as well as H-4*β*, are stable, while configurations at C-2, C-13, and C-14 positions vary. However, outcomes from nuclear Overhauser effect spectroscopy (NOESY) and ROESY experiments are often inconclusive because of the high flexibility of the 12-membered ring. Previously, Su et al. summarized empirical rules for deducing the relative configurations of C-2, C-13, and C-14 based on the chemical shifts of specific proximal carbons (CH_3_-16, C-4, C-13) [12]. When establishing the relative configurations of C-2, C-13, and C-14 for compounds **3***–***21**, the chemical shifts of CH_3_-16 and C-13 are indeed correlated to the orientations of CH_3_-16 and H-13. However, once C-7 was oxygenated, leading to the chemical shift of C-4 exceeding 45 ppm, the orientation of OAc-14 does not always align with the empirical rules, such as euphoscopoid E (*δ*_C-4_ 45.0), epieuphoscopin B (*δ*_C-4_ 45.3), and euphornin H (*δ*_C-4_ 46.2) [13]. Therefore, searching for additional methods dealing with the stereochemical questions of jatropha diterpenoids remains an urgent priority. We summarized some efficient ^1^H NMR spectroscopic rules for assigning the relative configurations of C-2, C-13, and C-14 in (7,8-*seco*)/3*α*-benzyloxy-jatropha-11*E*-ene derivatives, providing a valuable complement to ^13^C NMR spectroscopic rules [12]. Based on the previous ^13^C NMR rules and literature research, the configurations of compounds **3***–***21** containing a (7,8-*seco*)/3*α-*benzyloxy-jatropha-11*E*-ene were successfully established. The relative configurations of C-2, C-13, and C-14 can be fully and readily assigned by analyzing ^1^H NMR information, including coupling constants and chemical shifts of H-11 and H-12 (Figure 4, Table 4 and Table 5). First of all, coupling constants of H-2/H-3 and H-13/H-14 determine whether the orientations of H-2/H-3 and CH_3_-20/14-OAc are the same or not. For instance, the signal of H-3 appeared as t-like (*J* ≈ 3.0–5.0 Hz), indicating a 16*α*-CH_3_, while the signal of H-3 appeared as a doublet of doublets (*J* ≈ 8.0, 3.0–5.0 Hz) indicating a 16*β*-CH_3_. Similarly, if the ^3^*J*_H-13_, _H-14_ values range from 1.0 to 4.0 Hz, the orientation of CH_3_-20 and 14-OAc/OH are the same, whereas they should adopt a different orientation. Moreover, if the chemical shift of H-11 was smaller than that of H-12, the CH_3_-20 group was *α* oriented; otherwise, the CH_3_-20 should be *β* oriented (Table 4 and Table 5).

Theoretically, these correlations are attributable to steric interactions within the molecules. For all compounds, ^3^*J* values indicative of the dihedral angle between adjacent protons serve as a practical means to infer their orientation relationship. In these analogs with a 3*α*-benzyloxy-jatropha-11*E*-ene system, the C-13 position includes a methyl substituent (CH_3_-20), and the C-12 position contains olefinic hydrogen: When CH_3_-20 is *α*-oriented, the *gauche* relationship between 20*α*-CH_3_ and H-11 induces a significant upfield shift in H-11, attributable to the *γ*-*gauche* effect (Figure 4); conversely, when CH_3_-20 is *β*-oriented, the increased distance between 20*β*-CH_3_ and H-11 does not result in a steric interaction between these groups. The above correlations align with NMR data from the reported jatropha diterpenes, whose structures have been confirmed by X-ray analysis [11,12,13,19,20,21,22,23]. These empirical ^1^H NMR spectrum rules are also suitable for 7,8-*seco*-jatrophanes with similar signals (Figure 4). Therefore, according to the empirical rules summarized in this and previous research [12], determining the relative configurations of C-2, C-13, and C-14 in jatropha diterpenoids can be efficiently achieved by analyzing their ^1^H and ^13^C NMR spectra.

Given the empirical rules, the structure of euphorbiapene C (**13**) required a correction of the orientation of OAc-14 (14*β*-OAc other than 14*α*-OAc) for coupling constant between H-13 and H-14 (*J* = 8.7 Hz) and chemical shift of C-4 (44.8 ppm < 45 ppm) [19,20]. The remaining fourteen known compounds were identified as euphoheliosnoid A (**8**) [24], euphoscopins B (**9**)/C (**10**) [25], euphornins F (**11**) [26]/G (**12**) [19], helioscopianoid A (**14**) [23], euphornin K (**15**) [26], euphoscopin J (**16**) [20], euphoscopin D (**17**) [20], euphornin C (**18**) [20,26], helioscopianoid M (**19**) [23], 14*α*-acetoxy-3α-benzyloxy-7*α*,9*β*,15α-trihydroxy jatropha-5*E*,11*E*-diene (**20**) [27], secoheliosphane B (**21**) [11], *ent*-16*β*,17-dihydroxyatlsan-3-one (**22**) [15], and 2*α*-hydroxy helioscopinolide B (**23**) [28] by comparison of their ^1^H and ^13^C NMR data with those reported in the literature.

### 2.2. Biological Activity Results

In China, *E. helioscopia* is one of the main herbs to treat psoriasis [5,7,8,14], a skin disease characterized by the excessive proliferation of keratinocytes and lymphocytes. Therefore, compounds **1**–**23** were evaluated for their immunosuppressive activities against the proliferation of T/B lymphocyte cells and HaCaT cells (Human immortalized keratinocytes) in vitro. Preliminary results (Table 6) indicated that nearly all diterpenoids moderately inhibited concanavalin A (ConA)-induced T lymphocytes and/or lipopolysaccharide (LPS)-induced B lymphocytes proliferation (*c* 20.0 μM), as well as HaCaT cell proliferation (*c* 40.0 μM). Moreover, compounds **5** and **7** displayed significant immunosuppressive activities, with IC_50_ values of 17.6/10.2 and 6.7/11.5 μM (Table 7) against induced T and B cells. Compound **13** exhibited stronger anti-proliferative activities on HaCaT cells than the positive control, MTX (Table 8). Since compounds **7** and **21** in different jatropha-type families showed considerable immunosuppressive activity, they were also evaluated for their inhibitory effect by EdU experiments. Compared with the model or control controls, EdU experiments demonstrated that compounds **7**/**21** and **9**/**13** significantly inhibited T/B and HaCaT cells, respectively (Figure 5 and Figure 6) (Appendix A).

In addition, compounds **7** and **21** were also evaluated for their inhibitory effect on the secretion of cytokine interferon (IFN) *γ*, interleukin (IL) 2, and IL-17A of mouse splenocytes, as previously described [29]. Lymphocytes were seeded in 96-well plates and treated with compounds **7** and 21 at concentrations of 20, 10, 5, 2.5, and 0 μM for 48 h. Cell viability was analyzed using a cell counting kit-8 (CCK-8) assay [30]. Since there were no obvious differences in cell viability between **7**/**21** (*c* = 2.5 μM) and the control group, inhibitory activity against the secretion of cytokines was evaluated at a concentration of 2.5 μM and 1 μM (Figure 5A). As shown in Figure 5B, after being stimulated with ConA, the levels of IFN-γ/IL-2/IL-17A increased significantly in the model group via the control group (*p* < 0.0001); when dealt with compounds **7** and **21**, the secretions of IFN-*γ* (*p* < 0.0001) and IL-2 (*p* < 0.01) were significantly inhibited. Although compounds **7** and **21** reduced the secretion of IL-17A, there is no significant difference compared with the model group.

Nuclear factor-*κ*B (NF-*κ*B) is a family of heterodimeric proteins, including proteolytic processing of the p50 subunit as well as the P65 subunit. NF-*κ*B activation results in the expression of genes associated with cellular proliferation, differentiation, and survival [31]. Therefore, the most promising compound **13** for psoriasis was chosen to evaluate its molecular mechanisms associated with the NF-*κ*B pathway for the antiproliferative effect on HaCaT cells. As indicated in Figure 7, compound **13** attenuated the activation of NF-*κ*B in a concentration-dependent manner (10 and 5 μM). Molecular docking analysis between **13** and P65 revealed that (i) the oxygen at C-3 and the carbonyl group of acetoxy groups at C-7 and C-14 formed hydrogen bonds with the residues THR164, ARG73, and GLN142, which make the structure of complex P65-**13** stable; (ii) alkyl interaction was generated between 16-CH_3_ and the residue PRO172 (Figure 7). The binding energy of **13** with P65 is −7.3 kcal/mol. These results suggested that compound **13** inhibits the phosphorylation of NF-*κ*B p65 in HaCaT cells.

## 3. Discussion

The structural elucidation of natural products remains a significant challenge for chemists. Developing new methods using JBCA, examining the intensity of the nuclear overhauser effect, relying on quantum-chemical calculations or X-ray crystallographic analysis, and summarizing empirical rules of NMR data are effective procedures for determining the relative configurations of organic molecules [12,13,18,19]. Recently, the relationships between the relative configurations of C-2/C-13/C-14 and ^13^C NMR chemical shifts (C-16, C-4, C-13) were summarized in specific jatropha diterpenes [12]. However, these rules did not encompass jatropha-11*E*-ene systems with (7,8-*seco*) 3α-benzyloxy (or hydroxy) derivatives or these hydroxy-substituted at C-3/C-14/C-15. Therefore, this work further analyzed the correlations between ^3^*J* values/chemical shifts of the relevant protons/carbons and the relative configuration of C-2, C-13, and C-14 in A/B-type jatropha diterpenes based on the previous study (Table 4 and Table 5) [12].

Psoriasis is a common immune-mediated skin disease with unclear cellular and molecular mechanisms; however, T cell-mediated hyperproliferation of keratinocytes is the key feature of the pathophysiology of psoriasis. Immunosuppressants, vitamin A acid, methotrexate, and glucocorticoids are traditional and classical clinical medicines [2]. Moreover, *E. helioscopia* is the predominant ingredient in the Chinese ZeQi Powder Preparation for treating psoriasis [5,14]. Kv1.3 ion channel is predominant in activated effector memory T (T_EM_) cells to further stimulate T_EM_ cell proliferation and cytokine production [8]. Although some studies have found helioscopids A/O/euphohelioscoids A from *E. helioscopia* inhibited Kv1.3 activity in human embryonic kidney cells 293 (HEK293) cells [7,8], this activity only indirectly suggests the immunosuppressive potential of these compounds. Thus, more relevant bioassay experiments are needed to elucidate how and why compounds from *E. helioscopia* contribute to psoriasis treatment. Additionally, natural products often vary due to their different origins.

Lymphocytes, which are the most important immune cells, can be divided into T and B lymphocytes. During an immune response, T and B lymphocytes secrete various inflammatory factors (such as IL-2, IFN-*γ*, IL-17A, and so on) or antibodies to maintain internal environmental stability jointly. However, the overactivation of T or B lymphocytes can lead to various autoimmune diseases, such as psoriasis [32]. In vitro, the proliferation of T and B lymphocytes can be selectively stimulated by Con A and LPS, respectively. Meanwhile, the production level of cytokines and antibodies increases significantly when T and B lymphocytes are stimulated. Therefore, based on the specificity of these two mitogens and reference to the experimental model of splenic cell proliferation inhibition, this study evaluated the immunosuppressive activity of compounds from *E. helioscopia* using the previously established experimental model [33]. As depicted in Table 7 and Figure 5, most compounds displayed anti-proliferative effects on induced T and B cells. In addition, compounds **7** and **21** significantly reduced the secretions of IFN-*γ* (*p* < 0.0001) and IL-2 (*p* < 0.01). IL-2 and IFN-γ are closely associated with T cell proliferation [34]. Therefore, compounds **7** and **21** may exert anti-proliferative effects on T lymphocytes by reducing IFN-*γ* and IL-2 levels.

As previously reported, excessive proliferation and aberrant differentiation of keratinocytes are the main cellular events in psoriasis development [35]. Since the HaCaT cell line exhibits good stability and infinite proliferation ability, similar to the characteristics of the rapid proliferation of epidermal cells in psoriasis pathophysiological changes, it has been widely used in psoriasis models in vitro [35]. As shown in Table 8 and Figure 6, more than half of the compounds effectively inhibited HaCaT cells, particularly compound **13**. NF-*κ*B proteins are a family of transcription factors central to inflammation and immunity, and they also play crucial roles in development, cell growth, survival, proliferation, and various pathological conditions [36]. The mechanism of **13** was then explored, focusing on NF-*κ*B P65. Compound **13** may bind with NF-*κ*B P65, inhibiting its phosphorylation of NF-*κ*B P65 and thereby reducing the proliferation of HaCaT cells (Figure 7).

Compounds **5**/**6**, **8**/**9**, **11**/**12**, and **13**/**14** are four pairs of jatropha diterpenes differing only in their C-7 substitution (hydroxy or acetoxy). Analyzing the structure-activity relationship among these diterpenoid pairs, 7-OH is more beneficial than 7-OAc for the immunosuppressive activity of **5**, **9**, **12**, and **13**. However, 7-OAc is more effective than 7-OH in antiproliferative activity against HaCaT cells.

## 4. Materials and Methods

### 4.1. General Experimental Procedures

UV spectra were recorded on a Shimadzu UV-2401PC UV-visible recording spectrophotometer (Shimadzu, Kyoto, Japan). IR spectra (in CH_3_OH) were measured on a Bruker Tensor 27 spectrometer (Bruker, Karlsruhe, Germany). Rotations were measured using the APIV (Rudolph Research Analytical, Hackettstown, NJ, USA). A Chiral scan circular dichroism spectrometer (Applied Photophysics Ltd., Leatherhead, Surrey, UK) recorded the CD spectra. All the nuclear magnetic resonance (NMR) spectra were obtained on a Bruker Avance III 500 MHz spectrometer (Bruker Corporation, Karlsruhe, Germany). High-resolution electrospray ionization mass spectra (HRESIMS) were recorded using an AB Sciex Triple-TOF 6600 (AB SCIEX, Framingham, MA, USA). Sephadex LH-20 (Amersham Biosciences, Uppsala, Sweden) and silica gel (Qingdao Haiyang Chemical Co., Ltd., Qingdao, China) were used for column chromatography (CC). Medium-pressure liquid chromatography (MPLC) was performed on the Agela ODS flash column (C-18, 120 g, 40–60 mm, Tianjin, China) and the FL-H050G MPLC system (Agela Technologies, Tianjin, China). Preparative high-performance liquid chromatography (prep. HPLC) was performed on a SEP LC-52 equipped with a YMC-pack ODS-A column (dimensions 250 × i.d. 10 mm, particle size 5 μm, YMC, Tokyo, Japan) and a MWD UV detector (Separation Technology Co., Ltd., Beijing, China). All solvents used were analytical grade (TJshield Fine Chemicals Co., Ltd., Tianjin, China). All elution systems were described as volume ratio. Concanavalin A (ConA), lipopolysaccharide (LPS, *Escherichia coli* 055: B5), CCK-8, and RPMI 1640 medium were purchased from GibcoBRL, Life Technologies (Carlsbad, CA, USA). Fetal bovine serum (FBS) was purchased from HyClone Laboratories (Logan, UT, USA).

### 4.2. Plant Material

The whole plant, *E. helioscopia* L. (Euphorbiaceae), was collected from Kaifeng City, Henan Province, China, in May 2017 and identified by Prof. Cheng-Ming Dong (Henan University of Chinese Medicine), an expert in the taxonomic field of Chinese medicine. A voucher specimen (No. HZY201705) was deposited at the School of Pharmacy, Henan University of Chinese Medicine.

### 4.3. Extraction and Isolation

The whole air-dried and powdered *E. helioscopia* L. plant (6.0 kg) underwent a quadruple extraction with 95% ethanol. After evaporation under reduced pressure, the extract was partitioned between ethyl acetate and water four times, yielding a crude extract (420 g). The crude extract underwent normal column chromatography (CC) using a stepwise gradient of petroleum ether to acetone (from 10:1 to 1:1), resulting in three fractions (I–III). Fractions II (130 g) and III (120 g) were subjected to macroporous resin CC with gradient elution (EtOH/H_2_O: 20%, 40%, 60%, 80%, and 100%) each giving 14 (II_a_–II_n_) and 11 (III_a_–III_k_) subfractions. Subfractions II_d_, II_f_, and II_g_ were further separated by Sephadex LH-20 CC (CH_3_OH), leading to fractions IId1–IId4, IIf1–IIf4, and IIg1–IIg7. Then, fraction II_d3_ was fractioned on silica gel CC (isocratic elution of ether/acetone, 2:1). Finally, compound **10** (82.6 mg, *t*_R_ = 9.5 min) was purified from minor fraction II_d3a_ by HPLC (CH_3_CN/H_2_O, 70:30 to 100:0, 30 min, 5 mL/min). Fraction II_f1_ was separated on an MPLC equipped with an MCI CC (CH_3_OH/H_2_O: 20%, 40%, 60%, 80%, 100%) and 11 subfractions (II_f1a_-II_f1k_). Compound **22** (9.2 mg, *t*_R_ = 25.3 min) was isolated from II_f1f_ via HPLC (CH_3_CN/H_2_O, 50:50 to 80:20, 30 min, 5 mL/min). After preparation by HPLC (CH_3_CN/H_2_O, 50:50 to 70:30, 30 min, 5 mL/min) of fraction II_f1g_, **12** (11.7 mg, *t*_R_ = 24.8 min), **20** (2.6 mg, *t*_R_ = 27.8 min), **17** (4.2 mg, *t*_R_ = 28.3 min), and a mixture II_f1g9_. The mixture II_f1g9_ was purified by HPLC (CH_3_CN/H_2_O, 66:34 to 70:30, 30 min, 5 mL/min) again, then compounds **5** (14.2 mg, *t*_R_ = 18.8 min) and **16** (5.1 mg, t_R_ = 21.9 min) were isolated. Subfraction II_f1h_ was divided into six fractions (IIf1h1-IIf1h6) by HPLC (CH_3_CN/H_2_O, 63:37 to 70:30, 35 min, 5 mL/min). Respectively, compounds **15** (17.4 mg, *t*_R_ = 16.2 min; CH_3_CN/H_2_O, 68:32, 30 min, 5 mL/min) from subfraction II_f1h2_, and **14** (7.4 mg, *t*_R_ = 16.0 min; CH_3_CN/H_2_O, 66:34, 30 min, 5 mL/min) from subfraction IIf1h3 were obtained via HPLC once again. Fraction II_f1i_ gave compounds **6** (3.9 mg, *t*_R_ = 23.6 min) and **13** (1.5 mg, *t*_R_ = 39.0 min) via HPLC (CH_3_CN/H_2_O63:37 to 75:25, 40 min, 5 mL/min). Compounds **18** (2.1 mg, *t*_R_ = 25.0 min, CH_3_CN/H_2_O 42:58 to 100:0, 40 min, 4 mL/min) from subfraction II_f2_, **9** (4.6 mg, *t*_R_ = 16.0 min, CH_3_CN/H_2_O 55:45 to 80:20, 30 min, 5 mL/min) from subfraction II_f3_, and **3** (3.9 mg, *t*_R_ = 13.2 min, CH_3_CN/H_2_O 60:40 to 100:0, 35 min, 5 mL/min) from subfraction IIfg1 were obtained through HPLC. After HPLC (CH_3_CN/H_2_O 60:40 to 80:20, 35 min, 5 mL/min), fraction IIg2 afforded 12 components (IIg2a-IIg2l). Subsequently, compounds **7** (1.3 mg, *t*_R_ = 45.1 min CH_3_CN/H_2_O 46:54, 50 min, 5 mL/min) from subfraction II_g2a_, **21** (9.2 mg, *t*_R_ = 16.8 min; 60:40, 30 min, 5 mL/min) from subfraction II_g2b_, **19** (3.3 mg, *t*_R_ = 33.8 min CH_3_CN/H_2_O 55:45, 50 min, 5 mL/min) from subfraction II_g2c_, **11** (11.7 mg, *t*_R_ = 35.9 min; CH_3_CN/H_2_O 50:50, 50 min, 5 mL/min) from subfraction II_g2g_, and **4** (4.7 mg, *t*_R_ = 25.6 min; CH_3_CN/H_2_O 58:42, 50 min, 5 mL/min) from subfraction II_g2k_ via HPLC. Successively, components III_h_ and III_i_ were subjected to MPLC equipped with a C-18 column (CH_3_OH/H_2_O, 40%, 60%, 80%, and 100%), each affording four (III_h1_-III_h14_) and sixteen fractions (III_i1_-III_i16_). Fraction III_h1_ (CH_3_OH/H_2_O 48:52 to 100:0, 50 min, 10 mL/min) and III_i8_ (CH_3_OH/H_2_O 75:25 to 100:0, 40 min, 10 mL/min) gave six (III_h1a_-III_h1f_) and seven (III_i8a_-III_i8g_) fractions using HPLC. The compounds **1** (6.7 mg, *t*_R_ = 26.2 min) and **2** (7.9 mg, *t*_R_ = 27.2 min) were obtained from part III_h1e_ purified by HPLC (CH_3_CN/H_2_O 42:58 to 70:30, 30 min, 4 mL/min) while compounds **23** (3.9 mg, *t*_R_ = 12.6 min, CH_3_CN/H_2_O 50:50 to 80:20, 35 min, 4 mL/min) from subfraction III_i8d_, **8** (2.5 mg, *t*_R_ = 32.8 min, CH_3_CN/H_2_O 55:45 to 80:20, 35 min, 4 mL/min) from subfraction III_i8e_ were purified by HPLC.

### 4.4. Spectroscopic Data

*Ent*-2,3-*seco*-14-oxo-16-atisene-2,3-dioic acid (**1**): Amorphous powder, [α${]}_{D}^{25}$—34.7 (*c* 0.016, MeOH); UV (MeOH) *λ*_max_ (log*ε*): 295.0 (2.10), 200.0 nm (3.51); IR (MeOH) *ν*_max_/cm^−1^ 1701, 1721, 1636, 1530, 1214, 1161; ^1^H NMR (500 MHz CDCl_3_) data see Table 1; ^13^C NMR (125 MHz CDCl_3_) data see Table 1; HRESIMS *m*/*z* 371.1808 [M+Na]^+^ (calcd for C_20_H_28_O_5_Na, 371.1829).

*Ent*-2,3-*seco*-2,3-dicarboxy-abieta-8(14),13(15)-dien-16,12-olide (**2**): Amorphous powder, [α${]}_{D}^{25}$ + 130.1 (*c* 0.016, MeOH); UV (MeOH) *λ*_max_ (log*ε*): 275.0 (4.54), 205.0 nm (4.11); IR (MeOH) *ν*_max_/cm^−1^ 3394, 1735, 1674, 1639, 1029; ^1^H NMR (500 MHz CDCl_3_) data see Table 1; ^13^C NMR (125 MHz CDCl_3_) data see Table 1; HRESIMS *m*/*z* 385.1613 [M+Na]^+^ (calcd for C_20_H_26_O_6_Na, 385.1622).

Secoheliosphane C (**3**): Amorphous powder, [α${]}_{D}^{25}$ + 112.1 (*c* 0.02, MeOH), ^1^H NMR (500 MHz CDCl_3_) data see Table 2; ^13^C NMR (125 MHz CDCl_3_) data see Table 3; HRESIMS *m*/*z* 563.2564 [M+K]^+^ (calcd for C_31_H_40_O_8_K, 579.2354).

Secoheliosphane D (**4**): Colourless oil, [α${]}_{D}^{25}$ + 128.1 (*c* 0.02, MeOH); ^1^H NMR (500 MHz CDCl_3_) data see Table 2; ^13^C NMR (125 MHz CDCl_3_) data see Table 3; HRESIMS *m*/*z* 537.2459 [M+K]^+^ (calcd for C_29_H_38_O_7_K, 537.2449).

Heliosco-jatrophane A (**5**): Colourless oil, [α${]}_{D}^{25}$ + 98.6 (*c* 0.02, MeOH); ^1^H NMR (500 MHz CDCl_3_) data see Table 2; ^13^C NMR (125 MHz CDCl_3_) data see Table 3; HRESIMS *m*/*z* 563.2616 [M+Na]^+^ (calcd for C_31_H_40_O_8_Na, 563.2615.

Heliosco-jatrophane **B** (**6**): Amorphous powder, [α${]}_{D}^{25}$ + 51.3 (*c* 0.02, MeOH) ^1^H NMR (500 MHz CDCl_3_) data see Table 2; ^13^C NMR (125 MHz CDCl_3_) data see Table 3; HRESIMS *z m*/*z* 605.2714 [M+Na]^+^ (calcd for C_33_H_42_O_9_Na, 605.2721.

Heliosco-jatrophane **C** (**7**): Colourless oil, [α${]}_{D}^{25}$ + 100.1 (*c* 0.016, MeOH); ^1^H NMR (500 MHz CDCl_3_) data see Table 2; ^13^C NMR (125 MHz CDCl_3_) data see Table 3; HRESIMS *m*/*z* 521.2530 [M+Na]^+^ (calcd for C_29_H_38_O_7_Na, 521.2515).

### 4.5. Biological Activity Assays

Immunosuppressive activities assay and antiproliferative activity against HaCaT cells were performed according to the reported methods [33].

#### 4.5.1. Immunosuppressive Activities Assay

Female or male Balb/c mice (6–8 weeks old) were sacrificed, and their spleens were removed aseptically. A single-cell suspension was prepared by pressing the spleens against the bottom of the Petri dish with a 5 mL syringe plunger, and cell debris and clumps were removed. Erythrocytes were depleted with ammonium chloride buffer solution, and the mononuclear cell suspensions were maintained in RPMI 1640 media containing 10% fetal bovine serum, penicillin (100 U/mL), and streptomycin (100 μg/mL). Then the mononuclear cell suspensions were dispensed into 96-well plates (2 × 10^5^ cells/well), in the absence or presence of compounds (*c* = 20.0 μM), were stimulated with ConA (5 μg/mL) or LPS (15 μg/mL) to induce T cell or B cell proliferative responses, respectively. Dexamethasone (Dex, *c* = 2.0 μM) was used as a positive control. In the 96-well plates, the wells cultured with cells and ConA/LPS were assigned as the modal group (Mod), the wells containing cells were described as the control group (Con), and the wells containing culture medium were described as the blank group. These 96-well plates were maintained under a humidified 5% CO_2_ atmosphere at 37 °C for 48 h. 10 μL of CCK-8 was added to each well at the final 4–5 h of culture, and the absorbance (OD) values were measured with a microplate reader (SpectraMax iD3, Molecular Devices, LLC.) at 450 nm. Stimulation Index (SI) = (OD_sample_−OD_blank_)/(OD_Con_−OD_blank_); Inhibition rate = (1−SI_sample_/SI_Mod_) ×%. The IC_50_ value for each compound was calculated using the Reed and Muench method. All experiments were performed in triplicate.

#### 4.5.2. Cell Viability Assays of Lymphocytes

Similarly, the mononuclear cell suspensions of lymphocytes (2 × 10^5^ cells/well) were cultured within 96-well plates in the absence or presence of compounds **7** or **21** (*c* = 20.0, 10, 5, 2.5, and 0 μM). In the 96-well plates, the wells containing cells were described as the control group (Con), and the wells containing culture medium were described as the blank group. These 96-well plates were maintained under a humidified 5% CO_2_ atmosphere at 37 °C for 48 h. 10 μL of CCK-8 was added to each well at the final 5 h of culture, and the OD values were measured with a microplate reader at 450 nm. Cell vialibity (%) = (OD_sample_−OD_blank_)/(OD_Con_−OD_blank_) × 100%.

#### 4.5.3. Antiproliferative Activity against HaCaT Cells

The keratinocyte cell line was HaCaT (FH0186), bought from Fufeng Biology (Shanghai, China). HaCaT cells were cultured using the trypsin enzyme digesting technique and then passaged in vitro. The number of digested cells was counted by cell counter and were maintained in DMEM media containing 10% fetal bovine serum, penicillin (100 U/mL), and streptomycin (100 μg/mL) under a humidified 5% CO_2_ atmosphere at 37 °C. In brief, each well of a 96-well cell culture plate was seeded with 100 μL of cells (1 ×10^5^ cells/mL) and kept for 12 h for adherence and then added with test compounds (at a final concentration of 40 μM). The wells containing cells were described as the control group (Con), and the wells containing culture medium were described as the blank group. Methotrexate (MTX) was used as a positive control. After different concentrations of test compounds addition, each cell line was incubated for 48 h under a humidified 5% CO_2_ atmosphere at 37 °C. After the incubation, each well was treated with CCK-8 (10 μL), and incubation continued for 4 h at 37 °C. Then, the 96-well cell culture plates were subjected to a measure of optical density at 450 nm with a 96-well microplate reader. The IC_50_ value for each compound was calculated using the Reed and Muench method. All experiments were performed in triplicate.

#### 4.5.4. EdU Assay

EdU fluorescence labeling for cell proliferation of induced T/B and HaCaT cells was performed as previously reported [37,38]. EdU (the Click-iT™ EdU Flow Cytometry Assay Kit, APExBIO, Houston, TX, USA) was added 24 h (final concentration 50 μM, T/B cells) and 4 h (final concentration 10 μM, HaCaT cells) before harvesting the cells. For the Click reaction, cells were fixed with 100 μL of 4% paraformaldehyde for 15 min. Cells were washed again and incubated with 100 μL of saponin-based permeabilization buffer for 15 min. After additional washing, cells were incubated with 100 μL Click-iT reaction buffer for 1 h and washed again with 200 μL permeabilization buffer. All procedures were performed according to the manufacturer’s instructions.

#### 4.5.5. Cytokine Analysis by ELISA of Induced T Cells

The mononuclear cell suspensions (2 × 10^5^ cells/well) were cultured with ConA (5 μg/mL) in 96-well plates, and indicated concentrations of compounds were added simultaneously. After a 48-h culture period, cytokines in the supernatants were quantified using mouse IFN-*γ*, IL-2, and IL-17A ELISA kits (Mabtech, Stockholm, Sweden), following the manufacturer’s protocol [29].

#### 4.5.6. Immunofluorescence Protocol (Cell Climbing Slides)

HaCaT cells (2.0 × 10^5^ cells/well) were plated onto 6-well plates with glass coverslips and allowed to adhere overnight (12 h) before compound addition. After treatment with compound **13** (final concentration 10/5 μM) for 36 h, the supernatant was removed, and the cells were washed twice with PBS, fixed with 4% paraformaldehyde for 15 min, and then washed with PBS three times. After blocking for 1 h with 4% bovine serum albumin (BSA) at 37 °C, the supernatant was discarded, washed three times, and incubated with NF-*κ*B P65 and phosphorylated NF-*κ*B P65 antibodies (1:200 diluted at 4% BSA; Servicebio, Wuhan, China) overnight at 4 °C. The initial incubation was followed by another incubation with secondary antibodies (Cy3 conjugated Goat Anti-Rabbit IgG, 1:300 diluted in 4% BSA; AlexaFluor^®^488-conjugated Goat Anti-Mouse IgG, 1:400 diluted in 4% BSA) (Servicebio, Wuhan, China) for another 50 min at room temperature. Then, cells were counterstained with 4,6-diamino-2-phenylindole (DAPI) for 5 min after washing three times with PBS. After the PBS washing, fluorescent seal liquid (PBS) was added, and the plate was monitored under an imaging system (Pannoramic MIDI, 3Dhistech, Budapest, Hungary). The nucleus is blue-labeled with DAPI. Positive cells are green (phosphorylation NF-*κ*B P65 (P-P65) or red (NF-*κ*B P65). The immunofluorescence areas for each indicator were analyzed in Image J 1.46r.

#### 4.5.7. Statistics and Reproducibility

Data analyses were carried out using Prism 8.0, and One-way ANOVA was used to compare the differences between groups. LSD or Dunnett’s T3 test was used for pair comparison according to standard deviation values. *p* < 0.05 or *p* < 0.01 were considered to be statistically significant.

#### 4.5.8. Ethic Statement

The animal study (No. DWLL202003116) was reviewed and approved by the Animal Welfare Ethics Committee of the Henan University of Chinese Medicine.

### 4.6. Molecular Docking Studies

The crystal structure of the inducible nitric oxide synthase (P65) (PDB: 1NFI) was downloaded from RCSB PDB (http://www.rcsb.org/, accessed on 6 June 2024), the structure of **13** was drawn by ChemDraw 14.0, and the 3D structure file was transformed by Chem3D 14.0. Before docking, the water molecules on the receptors were removed, and polar hydrogen atoms, charge, and magnetic field were added. AutoDockTools 1.5.6 was used to process ligands and receptors, and AutoDock vina was used for molecular docking. The 3D and 2D diagrams of the best-scored binding pose were visualized by Discovery Studio Visualizer v20 (http://www.discoverystudio.net/ accessed on 6 June 2024).

## 5. Conclusions

Phytochemical investigation of *E. helioscopia* resulted in isolating 23 diterpenoids, including four rare novel *seco*-diterpenoids (**1**–**4**) and three new jatropha diterpenes (**5**–**7**). Based on the NMR data obtained in this study and the previously summarized configuration rules, as well as those with crystallographic structures reported in the literature, the correlations between ^3^*J* values/chemical shifts of the relevant protons and the relative configuration of C-2, C-13, and C-14 in jatropha diterpenes were considered.

Bioassay results showed that almost all diterpenoids, especially the jatropha-type ones, simultaneously inhibited the proliferation of induced T/B, HaCaT cells, and the secretion of IFN-*γ* and IL-2. Immunofluorescence results and molecular docking studies suggested that compound **13** contributed, at least partially, to its antiproliferative effect on HaCaT cells by inactivating NF-*κ*B through decreasing phosphorylation of P65. This work first explored the promising for psoriasis of jatropha diterpenoids from two aspects: T/B lymphocytes and HaCaT cells.

## Figures and Tables

**Figure 1 molecules-29-04104-f001:**
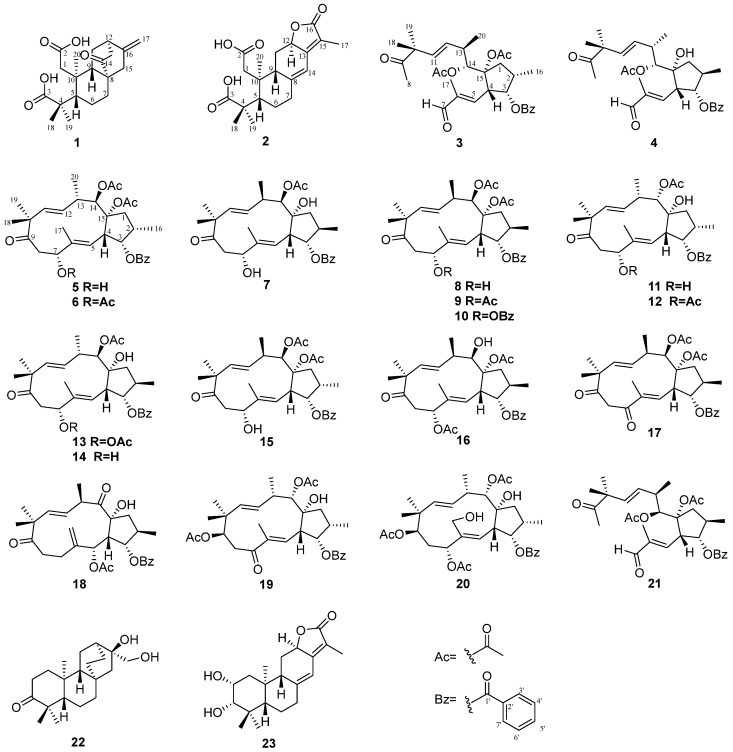
Structures of compounds **1**–**23** isolated from *E. helioscopia* L.

**Figure 2 molecules-29-04104-f002:**
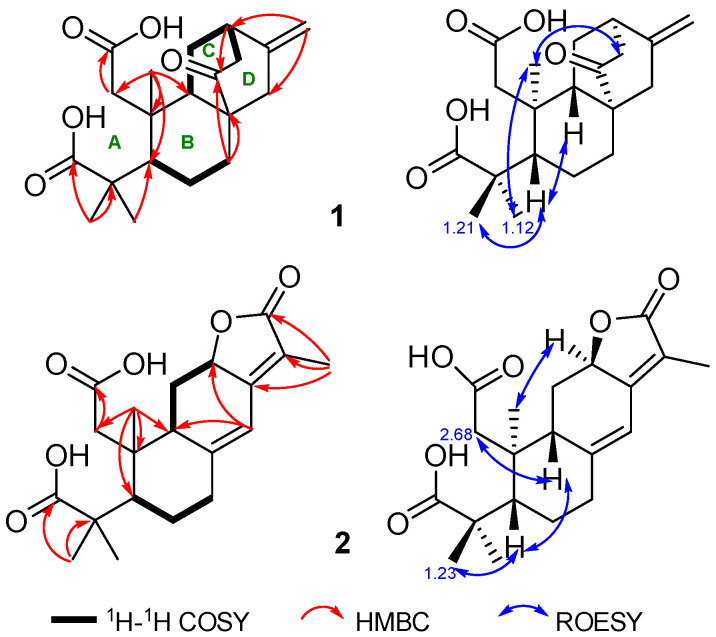
Key 2D NMR correlations of compounds **1** and **2** (four rings numbered as A–D).

**Figure 3 molecules-29-04104-f003:**
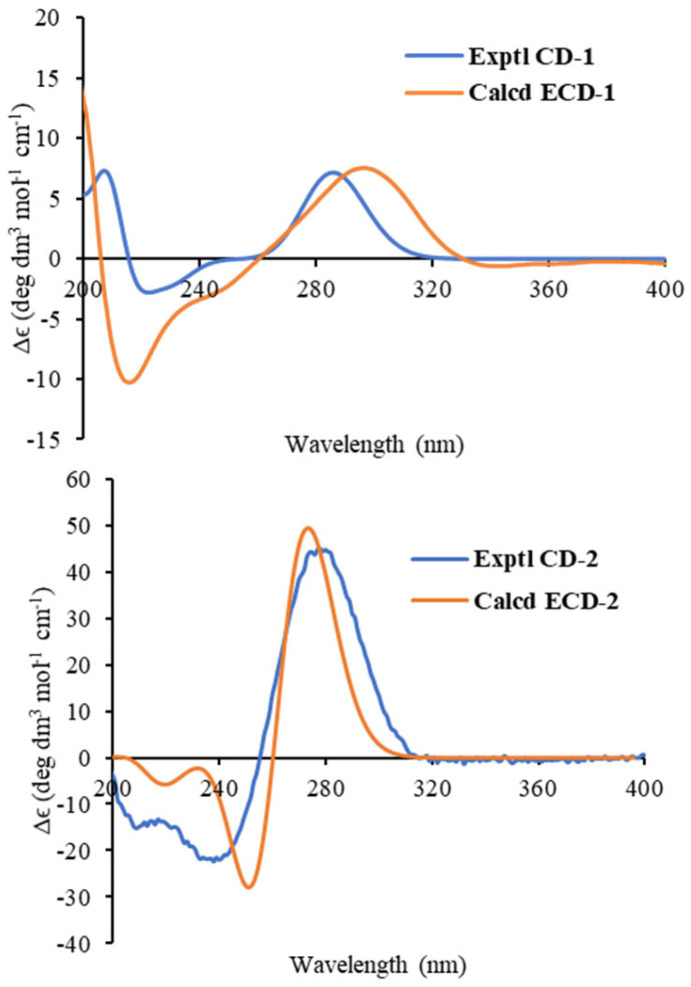
Experimental CD spectra and calculated ECD spectra of **1** (σ = 0.22 eV, UV shift—7 nm) and **2** (σ = 0.25 eV, UV shift—4 nm).

**Figure 4 molecules-29-04104-f004:**
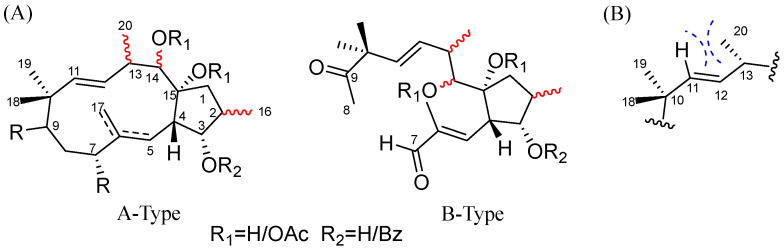
(**A**) Types A (3*α*-benzyloxy-jatropha-11*E*-ene) and B (8-*seco*-3*α*-benzyloxy-jatropha-11*E*-ene)**,** whose relative configurations of C-2, C-14, and C-13 can be determined by analysis 1D NMR data. (**B**) *γ*-Gauche relationship of 20*α*-CH_3_ with H-11.

**Figure 5 molecules-29-04104-f005:**
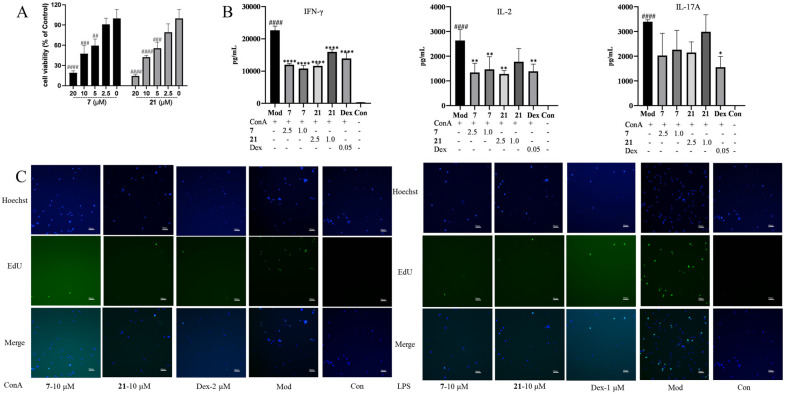
(**A**) Cell viability (% of Control) of compounds **7** and 21 in different concentrations (vs. Con ^####^ *p* < 0.0001, ^###^ *p* < 0.001, ^##^ *p* < 0.01). (**B**) Concentrations of IFN-γ/IL-2/IL-17A of induced T cells (Con A 5 μg/mL) in different groups (vs. Con ^####^ *p* < 0.0001; vs. Mod, **** *p* < 0.0001, ** *p* < 0.01, * *p* < 0.05). (**C**) Inhibitory activities of compounds **7** (*c* = 10 μM), **21** (*c* = 10 μM), and Dex (*c* = 2/1 μM) against induced T cells (Con A 5 μg/mL) and B (LPS, 15 μg/mL) measured by EdU.

**Figure 6 molecules-29-04104-f006:**
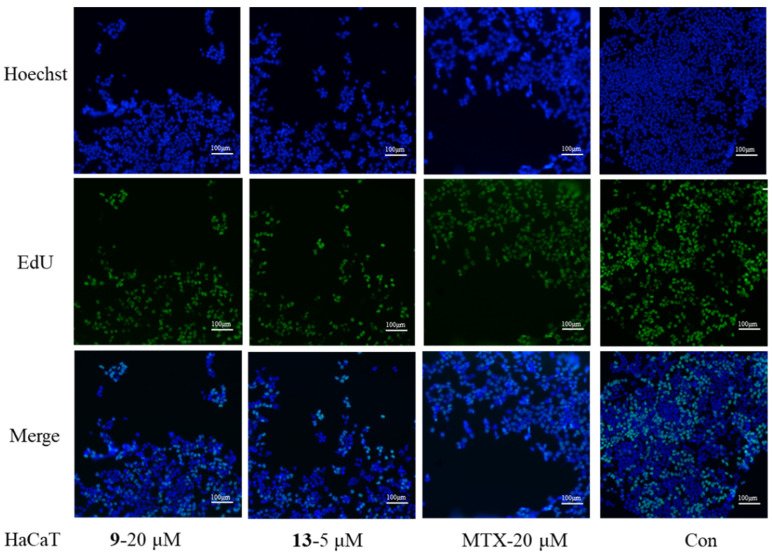
Inhibitory activities of compounds **9** (*c* = 20 μM), **13** (*c* = 5 μM), and MTX (*c* = 20 μM) against HaCaT cells measured by EdU.

**Figure 7 molecules-29-04104-f007:**
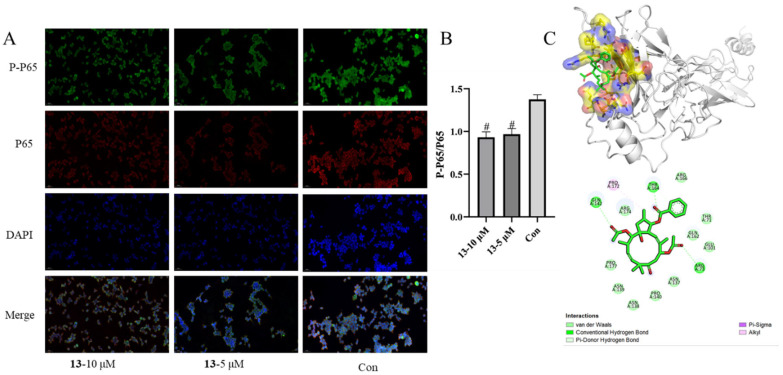
(**A**) Immunofluorescence images: Effects of compound **13** (*c* = 10/5 μM) on P-NF-*κ*B P65 (P-P65) and NF-*κ*B P65 (P65) expression levels in HaCaT cells. (**B**) Ratios of P-P65/P65 of HaCaT cells in different groups (vs. Con, ^#^ *p* < 0.05). (**C**) Molecular docking analysis for predicted lowest-energy binding mode of P65 in 3-dimensional and 2-dimensional figures with **13** (Affinity −7.3 Kcal/mol); for ligand, the carbon and oxygen are highlighted in green and red, respectively; for residents of proteins, the yellow, red and blue stand for C, O, and N, respectively.

**Table 1 molecules-29-04104-t001:** ^13^C (125 MHz) and ^1^H (500 MHz) NMR data of compounds **1** and **2** (*δ* in ppm, *J* in Hz, measured in CDCl_3_).

No.	1		2	
	*δ* _C_	*δ* _H_	*δ* _C_	*δ* _H_
1	39.0	2.79, d (19.7)	39.5	3.21, d (20.3)
		2.32, d (19.7)		2.68, d (20.3)
2	178.7		178.6	
3	187.5		186.9	
4	45.5. C		45.5	
5	48.3	2.54, d (12.4)	48.1	2.84, d (12.0)
6	19.5	1.61, m	23.3	2.03, m
		1.03, overlapped		1.53, overlapped
7	31.2	2.32, overlapped	36.1	2.55, d (13.8)
		0.98, overlapped		2.34, m, overlapped
8	47.7		151.1	
9	45.9	2.69 (m, overlap)	45.9	3.32, d (7.2)
10	41.3		43.6	
11	28.1	1.82, m	28.0	2.38, m
		1.56, m		1.59, m
12	38.6	2.74, m, overlapped	75.6	4.86 dd (13.0, 5.7)
13	44.7	2.31, m, overlapped	155.6	
14	217.2		114.5	6.30, s
15	42.9	2.27, m, overlapped	116.9	
		1.76, m, overlapped		
16	147.0		175.1	
17	107.3	4.88, s	8.3	1.83, s
		4.67, s		
18	29.5	1.21, s	29.5	1.23, s
19	21.7	1.12, s	21.3	1.20, s
20	17.4	0.78, s	21.0	1.01, s

**Table 2 molecules-29-04104-t002:** ^1^H (500 MHz) NMR data of compounds **3**–**7** (*δ* in ppm, measured in CDCl_3_, *J* in Hz).

No.	3	4	5	6	7
1	2.57, dd (12.5, 5.2) 2.12, overlapped	2.20, overlapped 1.53, dd (13.4, 11.1)	2.51, dd (14.0, 6.1) 1.95, t (14.0)	2.54, dd (13.6, 6.0) 1.99, d (13.6)	2.13, dd (14.5, 8.2) 1.51, dd (14.5, 8.6)
2	2.15, m	2.75, m	2.06, m	2.06, m	2.43, m
3	5.65, t-like (4.0)	5.20, dd (8.5, 5.7)	5.41, t-like (4.5)	5.45, t-like (4.4)	5.17, dd (6.8, 3.6)
4	3.61, dd (9.4, 4.0)	3.28, dd (9.8, 8.5)	3.26, dd (10.4, 4.5)	3.25, dd (9.9, 4.4)	3.16, dd (8.8, 3.6)
5	6.82, dd (9.3, 1.1)	6.77, brs (9.8, 1.3)	5.55, d (10.4)	5.69, d (9.9)	5.60, dd (8.8, 1.7)
7	9.36, s	9.32, s	4.16, t (7.5)	5.03, dd (8.1, 1.7)	4.39, dd (10.8, 4.5)
8	2.06, s	2.16, s	3.01, dd (14.5, 1.7) 2.44, dd (14.5, 6.9)	2.77, dd (13.6, 2.2) 2.86, dd (13.6, 8.4)	2.95, dd (15.4, 10.5) 2.69, dd (15.5, 4.6)
11	5.38, d (15.6)	5.58, d (15.9)	5.15, d (15.4)	5.16, d (15.4)	5.43, d (16.1)
12	5.27, dd (15.6, 8.8)	5.64, dd (15.9, 8.6)	5.25, dd (15.4, 8.8)	5.25, dd (15.4, 8.7)	5.15, dd (16.1, 9.0)
13	2.20, m	2.68, m	2.36, m	2.37, m	2.95, m
14	5.82, d (9.5)	4.94, d (3.1)	5.90, d (9.7)	5.93, d (9.9)	5.14, d (1.7)
16	0.96, d (7.0)	1.22, d (6.9)	0.92, d (6.6)	0.92, d (6.4)	1.08, d (5.9)
17	1.94, d (1.1)	1.68, d (1.4)	1.72, d (0.8)	1.81, br. s	1.81, d (1.5)
18	1.15, s	1.29, s	1.20, s	1.12, s	1.09, s
19	1.16, s	1.29, s	1.10, s	1.14, s	1.21, s
20	0.98, d (6.9)	0.99, d (6.9)	0.95, d (6.7)	0.97, d (6.7)	0.93, d (7.2)
3′, 7′	7.99, dd (7.5, 1.3)	7.93, dd (7.5, 1.3)	7.95, dd (7.5, 1.3)	7.97, d (7.5,1.3)	7.94, dd (7.5, 1.2)
4′, 6′	7.46, t (7.5)	7.39, t (7.5)	7.42, t (7.5)	7.43, t (7.5)	7.44, t (7.5)
5′	7.60, t (7.5)	7.52, t (7.5)	7.54, t (7.5)	7.55, t (7.5)	7.56, t (7.5)
OAc-14	2.17, s	1.90, s	2.16, s	2.14, s	2.16, s
OAc-15	2.05, s		2.14, s	2.20, s	
OAc-7				1.31, s	

**Table 3 molecules-29-04104-t003:** ^13^C NMR (125 MHz) data of compounds **3**–**7** (*δ* in ppm, measured in CDCl_3_).

No.	3	4	5	6	7
1	46.8	45.0	46.7	46.8	48.3
2	39.4	38.7	38.8	38.6	36.4
3	80.7	82.7	80.9	80.8	85.8
4	47.9	51.3	46.2	46.3	43.5
5	147.3	149.6	120.1	123	120.6
6	141.1	140.7	136.1	133.7	140.6
7	194.7	195	73.2	73.8	71.9
8	25.5	25.3	38.8	39.2	45.5
9	211.2	212.8	212.2	206.7	210.0
10	50.1	50.2	50.8	51.0	49.2
11	135.2	135.7	130.3	132.4	130.1
12	132.1	130.1	132.6	130.7	134.5
13	40.1	39.6	40.9	41.4	37.1
14	74.7	81.2	75.8	75.8	78.2
15	90.4	84.3	90.2	90.4	83.6
16	14.0	18.5	13.9	13.8	19.2
17	10.5	9.5	16.2	17.1	18.9
18	24.3	23.9	24.6	25.0	25.7
19	24	24.9	19.6	20.2	21.2
20	17.9	19.1	21.1	20.9	22.7
1′	165.9	166	166.1	165.4	166.3
2′	129.8	133.2	130.5	130.3	133.4
3′, 7′	129.6	129.6	129.6	129.6	129.5
4′, 6′	128.7	128.5	128.5	128.6	128.8
5′	133.5	131.7	133	133.1	133.5
OAc-14	169.3	170.6	169.8	170.0	170.7
	22.3	20.8	22.3	21.1	24.8
OAc-15	169.5		170.0	169.5	
	21.2		20.9	22.4	
OAc-7				170.2	
				20.2	

**Table 4 molecules-29-04104-t004:** Correlations between ^1^H NMR data signals and the relative configuration of C-2, C-14, and C-13 in types A or B.

^1^H NMR Data	Configurations	Suitable Type
H-3, t-like, *J* = 3.0–5.0 Hz	16*α*-CH_3_	A, B
H-3, dd, *J* ≈ 8.0, 3.0–5.0 Hz	16*β*-CH_3_	A, B
H-14, d, *J* = 1.0–4.0 Hz	CH_3_-20/OR-14 same orientation	A
H-14, d, *J* = 1.0–4.0 Hz	CH_3_-20/OR-14 same orientation	B
H-14, d, *J* > 8.0 Hz	CH_3_-20/OR-14 different orientation	A, B
*δ*_H-11_ > *δ*_H-12_	20*β*-CH_3_	A, B
*δ*_H-11_ < *δ*_H-12_	20*α*-CH_3_	A, B

**Table 5 molecules-29-04104-t005:** Correlations between ^13^C NMR data signals and the relative configuration of C-2, C-14, and C-13 in types A or B.

^13^C NMR Data	Configurations	Suitable Type
When OBz-3, *δ*_C-16_ < 15	16*α*-CH_3_	A, B
When OBz-3, *δ*_C-16_ > 15	16*β*-CH_3_	A, B
When OAc-14, *δ*_C-4_ < 45	14*β*-OAc	A
When OAc-14, *δ*_C-4_ > 45	14*α*-OAc or 14*β*-OAc	A
When OAc-14, *δ*_C-20_ > 22	CH_3_-20/OR-14 same orientation	A
when OAc-14, *δ*_C-20_ > 18.5	CH_3_-20/OR-14 same orientation	B
When OAc-15, *δ*_C-13_ < 40	20*β*-CH_3_	A, B

**Table 6 molecules-29-04104-t006:** Inhibition rate of compounds **1**–**23** (*c* 20.0 μM) and Dex (*c* 2.0 μM) against the ConA-induced proliferation of T and/or LPS-induced proliferation of B cells.

Compound	Inhibition Rate (%)		Compound	Inhibition Rate (%)	
	T Cells	B Cells		T Cells	B Cells
**1**	23.7 ± 4.9	<10	**13**	15.0 ± 2.7	27.2 ± 4.5
**2**	23.7 ± 6.8	<10	**14**	46.5 ± 8.3	28.2 ± 4.8
**3**	34.5 ± 5.4	20.0 ± 9.5	**15**	25.4 ± 3.5	17.1 ± 7.3
**4**	23.8 ± 4.5	26.6 ± 9.7	**16**	44.4 ± 8.6	26.6 ± 5.4
**5**	58.2 ± 1.4	64.9 ± 4.6	**17**	35.6 ± 1.1	<10
**6**	34.8 ± 7.4	<10	**18**	37.5 ± 5.6	33.5 ± 2.8
**7**	83.8 ± 2.5	92.4 ± 9.6	**19**	42.5 ± 7.5	48.3 ± 8.8
**8**	49.8 ± 6.9	43.1 ± 8.8	**20**	48.5 ± 7.1	46.6 ± 1.7
**9**	32.2 ± 2.9	27.1 ± 2.4	**21**	49.8 ± 7.8	48.2 ± 6.4
**10**	48.4 ± 3.2	48.7 ± 5.7	**22**	<10	<10
**11**	46.0 ± 3.2	17.6 ± 2.4	**23**	33.0 ± 8.9	<10
**12**	34.0 ± 3.3	14.2 ± 4.6	Dex	46.7 ± 4.3	79.3 ± 7.7

**Table 7 molecules-29-04104-t007:** IC_50_ values of compounds **5** and **7** against the induced proliferation of T cells (ConA) and B cells (LPS).

Compound	IC_50_ (μM)	
	T Cells	B Cells
**5**	17.6 ± 2.7	10.3 ± 1.3
**7**	6.7 ± 1.8	11.4 ± 1.5
Dex	1.6 ± 0.3	0.8 ± 0.05

**Table 8 molecules-29-04104-t008:** IC_50_ values of active compounds against the proliferation of HaCaT cells.

Compound	IC_50_ (μM)	Compound	IC_50_ (μM)
**7**	31.0 ± 3.7	**16**	25.9 ± 2.4
**8**	26.0 ± 2.2	**17**	31.5 ± 3.3
**9**	19.7 ± 2.0	**18**	29.8 ± 2.9
**10**	31.3 ± 1.7	**21**	28.5 ± 3.0
**12**	23.7 ± 2.5	**22**	29.6 ± 2.3
**13**	6.9 ± 0.8	**23**	30.4 ± 3.0
**15**	23.4 ± 1.6	MTX	18.1 ± 2.1

## Data Availability

The original contributions presented in the study are included in the article/Appendix A; further inquiries can be directed to the corresponding author.

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
