# Peer review of "Diterpenoids with Potent Anti-Psoriasis Activity from Euphorbia helioscopia L."

_molecules, 2024, doi:10.3390/molecules29174104_

Round 1

Reviewer 1 Report

Comments and Suggestions for Authors

The manuscript submitted by Zhen-Zhu Zhao et al., titled “Diterpenoids with Potent Anti-Psoriasis Activity from Euphorbia helioscopia L.”, is clear and well-written. However, to be considered for publication in the Molecules Journal, the authors must revise the manuscript in accordance with the suggestions made in the comments below.

  1. I find that the beginning of the abstract is missing a brief, one-sentence introduction that provides background information and presents the purpose of the research.
  2. The authors should highlight their contribution to existing research in the Results and Discussion section, and cite more research that has dealt with this topic. It would be also advisable to separate the Results and Discussion sections.
  3. Figure 7C has very poor resolution; the letters should be larger.
  4. The ’Experimental section’ should be renamed to ’Materials and Methods’.
  5. According to the guidelines of the Molecules Journal, the Conclusion section must follow the Materials and Methods section.
  6. Future viewpoints need to be covered in the Conclusion as well. The value of the research should be highlighted by the author.
  7. Section 4.4. Spectroscopic Data, would be more effectively summarized either in a table or in complete sentences.
  8. Please note that the section following 4.5.6 should not be numbered as 4.5.8.

Author Response

Dear editors and reviewers,

Thank you very much for your comments and professional advice. These opinions help to improve the academic rigor of our article. Based on your professional suggestions, we have corrected the revised manuscript's modifications. We hope our work can be improved again. Furthermore, we would like to show the details as follows:

Reviewer 1

The manuscript submitted by Zhen-Zhu Zhao et al., titled “Diterpenoids with Potent Anti-Psoriasis Activity from Euphorbia helioscopia L.”, is clear and well-written. However, to be considered for publication in the Molecules Journal, the authors must revise the manuscript in accordance with the suggestions made in the comments below.

I find that the beginning of the abstract is missing a brief, one-sentence introduction that provides background information and presents the purpose of the research.

Answer: Thank you for pointing this out. We agree with this comment. Therefore, we have added sentences in the abstract to provide background information and presents the purpose of the research. Please see the highlights lines of 13 and 14 in the revised manuscript.

The authors should highlight their contribution to existing research in the Results and Discussion section, and cite more research that has dealt with this topic. It would be also advisable to separate the Results and Discussion sections.

Answer: Agree. We have separated the Results and Discussion sections and highlighted our contribution in the Discussion section. Please see the highlights of the Discussion section in the revised manuscript.

Figure 7C has very poor resolution; the letters should be larger.

Answer: Thank you for pointing this out. We agree with this comment. Therefore, we have changed 7C and adjusted the letter.

The ’Experimental section’ should be renamed to ’Materials and Methods’.

Answer: Agree. The “Experimental section” has been renamed to “Materials and Methods”.

According to the guidelines of the Molecules Journal, the Conclusion section must follow the Materials and Methods section.

Answer: Thank you for pointing this out. We agree with this comment. Now, the Conclusion section follows the Materials and Methods section.

Future viewpoints need to be covered in the Conclusion as well. The value of the research should be highlighted by the author.

Answer: Thank you for pointing this out. We agree with this comment. Therefore, Future viewpoints need to be covered in the Discussion and Conclusion. The value of the research has been highlighted in the Discussion as well.

Section 4.4. Spectroscopic Data, would be more effectively summarized either in a table or in complete sentences.

Answer: Thank you. Spectroscopic Data was described in the most common pattern. The mistakes have been revised.

Please note that the section following 4.5.6 should not be numbered as 4.5.8.

Answer: Thank you for pointing this out. The section numbered 4.5.8 incorrectly has been corrected.

Reviewer 2 Report

Comments and Suggestions for Authors

This manuscript presents a comprehensive study on the isolation and structural elucidation of diterpenoids from Euphorbia helioscopia L., along with their evaluation for anti-psoriasis activity. The work is well-designed, and the data presented are extensive. However, there are a few aspects that could be improved to enhance the clarity, impact, and reproducibility of the study. My suggestions are as follows:

1. Abstract: Including more detail about the “bioassay” will enhance understanding. Briefly mention the type of bioassay conducted and the main findings related to the anti-psoriasis activity.

2. Figures: Figures 3, 5, and 7 lack titles, and Figure 6 lacks a figure legend. The abbreviations "M" and "Y" need to be clarified in the figure legends. Additionally, labels such as “7-10 µM” are confusing. Clearly indicate what these labels represent and ensure consistency in units and labeling across figures.

3. Results Section: The description of Figures 5 and 6 is currently insufficient, with only one sentence provided. Include a more detailed explanation of these figures to help readers understand the data presented. Discuss the significance of the findings shown in these figures and how they contribute to the overall conclusions of the study.

Comments on the Quality of English Language

The English language is fine in most parts, except in the figure legends where the language needs improvement.

Author Response

This manuscript presents a comprehensive study on the isolation and structural elucidation of diterpenoids from Euphorbia helioscopia L., along with their evaluation for anti-psoriasis activity. The work is well-designed, and the data presented are extensive. However, there are a few aspects that could be improved to enhance the clarity, impact, and reproducibility of the study. My suggestions are as follows:

  1. Abstract: Including more detail about the “bioassay” will enhance understanding. Briefly mention the type of bioassay conducted and the main findings related to the anti-psoriasis activity.

Answer: Thank you for pointing this out. We agree with this comment. Bioassay has been rewritten to give readers a better understanding. Please see the highlights in the section numbered 4.5..

  1. Figures: Figures 3, 5, and 7 lack titles, and Figure 6 lacks a figure legend. The abbreviations "M" and "Y" need to be clarified in the figure legends. Additionally, labels such as “7-10 µM” are confusing. Clearly indicate what these labels represent and ensure consistency in units and labeling across figures.

Answer: Thank you. Titles and legends of figures have been added. The abbreviations "M" and "Y" have been changed to “Mod” and “Con”. Additionally, labels such as “7-10 µM” have been explained in the legends. Please see the highlights in these figures.

  1. Results Section: The description of Figures 5 and 6 is currently insufficient, with only one sentence provided. Include a more detailed explanation of these figures to help readers understand the data presented. Discuss the significance of the findings shown in these figures and how they contribute to the overall conclusions of the study.

Answer: Agree. We have added more descriptions to explain Figures 5 and 6. An extra Discussion section has been supplied as well.

  1. Comments on the Quality of English Language

The English language is fine in most parts, except in the figure legends where the language needs improvement.

Answer: Thank you, we have improved the description of some figure legends.

Reviewer 3 Report

Comments and Suggestions for Authors

The manuscript titled “Diterpenoids with Potent Anti-Psoriasis Activity from Euphorbia helioscopia L.” is of great significance to isolate and identify anti-psoriasis compounds from Euphorbia helioscopia L.. In this manuscript, the methods and the results presentation were well organized. And the data interpretation was fairly clarity. However, the article needs to be revised before publication. 1. In this paper, all the pictures of the fluorescence microscope are not very clear, so that the pictures cannot show the exact experimental results. These must be promoted or improved. 2. In “Introdction section”, the author claims that some of Euphorbia genus are used as folk medicines in China to treat skin diseases (like psoriasis). Please supplement more references to discuss the phytochemical studies of Euphorbia genus in the treatment of psoriasis. 3. Figure 5A revealed that compounds 7 and 21 showed significant inhibitory activity against the secretion of IFN-γ and IL-2. I think this conclusion is unreasonable. These compounds significantly inhibited the cell proliferation of T/B cells and HaCaT cells. In figure 5A, the unit of IFN-γ/IL-2/IL-17A concentrations is pg/mL. The results in the figure 5A may be caused by a decrease in the number of cells, rather than an inhibitory effect on the secretion of cytokines. Please explain this and modify it in the text. 4. In all the pictures of the fluorescence microscope, there is a group named “Y”. What drug treatment this group is, need to be marked in the text, or using a common representation name. 5. Are new compounds isolated from this plant?

6. What is the structure-activity relationship of compounds? Please add in the text.

Author Response

The manuscript titled “Diterpenoids with Potent Anti-Psoriasis Activity from Euphorbia helioscopia L.” is of great significance to isolate and identify anti-psoriasis compounds from Euphorbia helioscopia L.. In this manuscript, the methods and the results presentation were well organized. And the data interpretation was fairly clarity. However, the article needs to be revised before publication. 

  1. In this paper, all the pictures of the fluorescence microscope are not very clear, so that the pictures cannot show the exact experimental results. These must be promoted or improved. 

Answer: Thank you for pointing this out. We agree with this comment. Therefore, we have enlarged the pictures of the fluorescence microscope.

  1. In “Introdction section”, the author claims that some of Euphorbiagenus are used as folk medicines in China to treat skin diseases (like psoriasis). Please supplement more references to discuss the phytochemical studies of Euphorbiagenus in the treatment of psoriasis. 

Answer: Agree. We have, accordingly, cited more relevant references to to discuss the phytochemical studies of Euphorbia genus in the treatment of psoriasis. Please see the highlights in the “Introduction section”.

  1. Figure 5A revealed that compounds 7 and 21 showed significant inhibitory activity against the secretion of IFN-γ and IL-2. I think this conclusion is unreasonable. These compounds significantly inhibited the cell proliferation of T/B cells and HaCaT cells. In figure 5A, the unit of IFN-γ/IL-2/IL-17A concentrations is pg/mL. The results in the figure 5A may be caused by a decrease in the number of cells, rather than an inhibitory effect on the secretion of cytokines. Please explain this and modify it in the text.

Answer: Thank you for pointing this out. We agree with this comment. In fact, we did the experiments to test the cell validities of compounds 7 and 21. In the revised manuscript, we have supplied the cell validity in Figure 5. Since there were no obvious differences in cell viability between 7/21 (c = 2.5 μM) and the control group, inhibitory activity against the secretion of cytokines was evaluated at a concentration of 2.5 μM and 1 μM (Figure 5A)

  1. In all the pictures of the fluorescence microscope, there is a group named “Y”. What drug treatment this group is, need to be marked in the text, or using a common representation name.

Answer: Thank you. The abbreviations "M" and "Y" have been changed to “Mod” and “Con”. Additionally, labels such as “7-10 µM” have been explained in the legends. Please see the highlights in these figures.

  1. Are new compounds isolated from this plant?

Answer: Seven new compounds were isolated from this plant, which were described in the abstract and conclusion sections.

  1. What is the structure-activity relationship of compounds? Please add in the text.

Answer: Thank you for pointing this out. Structure-activity relationship of compounds has been added in the Discussion section.

Reviewer 4 Report

Comments and Suggestions for Authors

The study presents the isolation of twenty-three diterpenoids from the whole herb of Euphorbia helioscopia L. and evaluates their bioactivity. While the research is promising, several aspects of the results and discussion require clarification and improvement. Below are detailed comments, concerns, and questions:

1.     Figure 3 Y-axis: Please verify the labeling and units on the Y-axis of Figure 3.

2.     Table in Figure 4: The current format of the table within Figure 4 is confusing. Consider revising it for better readability or separating it into a distinct table with an appropriate caption.

3.     Abbreviations: Ensure that all abbreviations are defined upon their first occurrence in the text. For example, provide the full names for ROESY, LPS, MTX, and Dex.

4.     Figure 5(B) and Figure 6: Clarify which compound was used in the inhibitory activity assay. Specify whether it was Compound 7, Compound 21, or a mixture of both. Provide additional information and discussion on this aspect.

5.     Bioactivity Discussion: Discuss the bioactivity of the compounds from Euphorbia helioscopia L. in terms of their correlation with anti-psoriasis activity. Provide a more detailed analysis of how each compound contributes to the observed effects.

6.     Spectroscopic Results: The methods section mentions the use of UV spectra, IR spectra, and rotational data. However, the corresponding results are missing. Include these results to provide comprehensive characterization of the compounds.

7.     Ethical Approval for Animal Study: Animal studies must have ethical approval. Include the ethical code and the approval details from the relevant animal ethics committee.

8.     CCK-8 Assay Method: Provide a detailed explanation of the CCK-8 assay method used in the study. This will ensure that the methodology is clear and reproducible.

9.     Cytotoxicity vs. Proliferation Evaluation: Review and clarify whether your method is intended to evaluate cytotoxicity or cell proliferation. Ensure that the description in the methods section accurately reflects the objective of the assay.

Author Response

The study presents the isolation of twenty-three diterpenoids from the whole herb of Euphorbia helioscopia L. and evaluates their bioactivity. While the research is promising, several aspects of the results and discussion require clarification and improvement. Below are detailed comments, concerns, and questions:

  1. Figure 3 Y-axis: Please verify the labeling and units on the Y-axis of Figure 3.

Answer: Thank you for pointing this out. Labeling and units on the Y-axis of Figure 3 have been added.

  1. Table in Figure 4: The current format of the table within Figure 4 is confusing. Consider revising it for better readability or separating it into a distinct table with an appropriate caption.

Answer: Agree. We have, accordingly, separated the table from Figure 4.

  1. Abbreviations: Ensure that all abbreviations are defined upon their first occurrence in the text. For example, provide the full names for ROESY, LPS, MTX, and Dex.

Answer: Thank you, we have checked the manuscript to ensure that all abbreviations are defined upon their first occurrence.

  1. Figure 5(B) and Figure 6: Clarify which compound was used in the inhibitory activity assay. Specify whether it was Compound 7, Compound 21, or a mixture of both. Provide additional information and discussion on this aspect.

Answer: Thank you. compound 7 and compound 21 were used separately, not as a mixture. The legend has been rewritten to clear that.

  1. Bioactivity Discussion: Discuss the bioactivity of the compounds from Euphorbia helioscopia L.in terms of their correlation with anti-psoriasis activity. Provide a more detailed analysis of how each compound contributes to the observed effects.

Answer: Thank you for pointing this out. A more detailed analysis of how the compounds contribute to the observed effects has been added in the discussion section.

  1. Spectroscopic Results: The methods section mentions the use of UV spectra, IR spectra, and rotational data. However, the corresponding results are missing. Include these results to provide comprehensive characterization of the compounds.

Answer: Thank you. For novel compounds 1 and 2, UV, IR spectra and rotational data were included. Rotational data were included for all new compounds.

  1. Ethical Approval for Animal Study: Animal studies must have ethical approval. Include the ethical code and the approval details from the relevant animal ethics committee.

Answer: Thank you for pointing this out. We agree with this comment. Therefore, Ethical Approval for Animal Study has been provided in the Ethic Statement section in the revised manuscript.

  1. CCK-8 Assay Method: Provide a detailed explanation of the CCK-8 assay method used in the study. This will ensure that the methodology is clear and reproducible.

Answer: Thank you.  We cited a reference to explain CCK-8 Assay Method [30.He, H.; Cao, L.; Wang, Z.; Wang, Z.; Miao, J.; Li, X. M.; Miao, M. Sinomenine relieves airway remodeling by inhibiting epithelial-mesenchymal transition through downregulating TGF-β1 and smad3 expression in vitro and in vivo. Front Immunol. 2021, 12, 736479. DOI:10.3389/fimmu.2021.736479].

  1. Cytotoxicity vs. Proliferation Evaluation: Review and clarify whether your method is intended to evaluate cytotoxicity or cell proliferation. Ensure that the description in the methods section accurately reflects the objective of the assay.

Answer: Thank you for pointing this out. We agree with this comment. Therefore, we have changed “cytotoxicity” to “anti-proliferation”. Cell Counting Kit-8 (CCK-8) can reflect the cell validity. Therefore, by comparing the model group or control group, we can learn whether the cell validity is high or low, which in turn reflects whether the compounds showed “anti-proliferation” or “cytotoxicity”. Sometimes, “anti-proliferation” and “cytotoxicity” express the same meaning.

Reviewer 5 Report

Comments and Suggestions for Authors

The current manuscript represents the diterpenoids analysis of Euphorbia helioscopia L., which has potent anti-psoriasis activity. The study's objective is well-defined and the references are up to date. However, the conclusion requires improvement to clearly demonstrate the significance of the study results. Although the concept is acceptable, there are some specific comments outlined below:

1. In the introduction section:

- In line number 53, “To find more diterpenoids with promising activities for psoriasis, E. helioscopia L. was selected to investigate in this study, resulting in the isolation of diterpenoids.” must be “To find more diterpenoids with promising activities for psoriasis, E. helioscopia L. was selected to be investigated in this study, resulted in the isolation of diterpenoids”.

2. In the results section:

- In line 264, “In China, E. helioscopia is one of the main herbs to treat Psoriasis”, please add a reference for this sentence.

- In line 270, “Moreover, compounds 5 and 7 displayed significant immunosuppressive activities, with IC50 values of 17.6/10.2, and 6.7/11.5 μM”, please clarify this sentence to the reader, IC50 against what cells?

- What do Dex and MTX refer to? Authors must write the full name beside the abbreviation the first time in the manuscript before using the acronym.

- In Table 4, T and B refer to what? Please add a note after the table for these T and B.

- In Figure 5, what does Y refer to?

- In line 298, “Therefore, the most promising compound 13 was chosen to evaluate its molecular mechanisms” The most promising compound for what?

Author Response

The current manuscript represents the diterpenoids analysis of Euphorbia helioscopia L., which has potent anti-psoriasis activity. The study's objective is well-defined and the references are up to date. However, the conclusion requires improvement to clearly demonstrate the significance of the study results. Although the concept is acceptable, there are some specific comments outlined below:

  1. In the introduction section:

- In line number 53, “To find more diterpenoids with promising activities for psoriasis, E. helioscopia L. was selected to investigate in this study, resulting in the isolation of diterpenoids.” must be “To find more diterpenoids with promising activities for psoriasis, E. helioscopia L. was selected to be investigated in this study, resulted in the isolation of diterpenoids”.

Answer: Thank you for pointing this out. we have revised this sentence. Please see highlights line 67.

  1. In the results section:

- In line 264, “In China, E. helioscopia is one of the main herbs to treat Psoriasis”, please add a reference for this sentence.

Answer: Agree. We have added some references for this sentence. Please see highlights line 282.

- In line 270, “Moreover, compounds 5 and 7 displayed significant immunosuppressive activities, with IC50 values of 17.6/10.2, and 6.7/11.5 μM”, please clarify this sentence to the reader, IC50 against what cells?

Answer: Thank you for pointing this out. We agree with this comment. Therefore, we have completed the descriptions as “IC50 values of 17.6/10.2, and 6.7/11.5 μM (Table 7) against induced T and B cells.”. Please see highlights line 290.

- What do Dex and MTX refer to? Authors must write the full name beside the abbreviation the first time in the manuscript before using the acronym.

Answer: Agree. full name of Dex and MTX were provided in the Biological activity section. Please see lines 512 and 541.

- In Table 4, T and B refer to what? Please add a note after the table for these T and B.

Answer: Thank you for pointing this out. T and B refer to T and B cells. Full names are given in Table 6.

- In Figure 5, what does Y refer to?

Answer: Thank you for pointing this out. The abbreviations "M" and "Y" have been changed to “Mod” and “Con”.

- In line 298, “Therefore, the most promising compound 13 was chosen to evaluate its molecular mechanisms” The most promising compound for what?

Answer: Thank you. This sentence has been corrected as “the most promising compound 13 for psoriasis was…”

Round 2

Reviewer 4 Report

Comments and Suggestions for Authors

-